# Response of Potted Citrus Trees Subjected to Water Deficit Irrigation with the Application of Superabsorbent Polyacrylamide Polymers

Daniela Cea, Claudia Bonomelli, Johanna Mártiz and Pilar M. Gil *

Departamento de Fruticultura y Enología, Facultad de Agronomía e Ingeniería Forestal, Pontificia Universidad Católica de Chile, Vicuña Mackenna 4860, Macul, Santiago 7820436, Chile; dfcea@uc.cl (D.C.); cbonomel@uc.cl (C.B.); jmartiz@uc.cl (J.M.)
* Correspondence: pmgil@uc.cl

**Abstract:** Searching for new strategies to mitigate the effects of low water availability for citrus production, a study was carried out on potted mandarin cv. W. Murcott, with the objective of evaluating the physiological and growth response of the plants to polyacrylamide gel application in the substrate in water restriction conditions. The following treatments were evaluated, T0 (control) with 100% ETc water replenishment, T1 with 50% ETc water replenishment, and T2 with 50% ETc water replenishment plus the application of polyacrylamide polymers to the substrate. Temperature and water volumetric content ($\theta$: $m^3$ $m^{-3}$) were evaluated in the substrate. Plant water-status parameters such as stem water potential (SWP), stomatal conductance (gs), and chlorophyll fluorescence (Fv/Fm), as well as biomass, nutrients levels, and proline biosynthesis were measured in the plants in response to the treatments. The results showed that the substrate moisture for T2 was kept significantly higher than T0 and T1, despite receiving the same irrigation rate as T1 and a half of T0; however, this higher moisture availability in the substrate of T2 was not reflected in the plant's water status or growth. On the contrary, the T2 plants showed responses such as lower total biomass, lower vegetative development, and lower root biomass, as well as a higher concentration of proline in the root. According to these results, it is concluded that polymers such as polyacrylamide sodium allow the retention of water in the substrate, but do not necessarily release that water for plants, probably because that moisture is kept in the hydrogel and not released to the substrate media or the roots, or if released, in this case, this occurs with an increase in the concentration of sodium available to the plants, which could lead the citrus crop to a worse situation of water and/or osmotic stress.

**Keywords:** hydrogel; irrigation deficit; mandarin; proline

## 1. Introduction

Drought is one of the major limiting factors for crop yields and causes important economic losses to the farmers. There is a constant search for tools to allow efficiency in the use of available water resources, save water, and increase crop productivity per unit of water used [1]. Climate change is threatening fruit production in several countries, mostly in Mediterranean countries where irrigation is necessary for agricultural production [1]. Under this context, several techniques and/or technologies have been evaluated in soil and/or plants in searching for strategies to deal with conditions of water shortage, without significantly affecting the yield and quality of fruit production.

In response to water stress, some tree species have a series of morphophysiological and physiological mechanisms that allow crop development despite stress conditions [2]. In citrus species, some forms of tolerance to water stress are based on increasing tissue elasticity, stomatal regulation [3], an adequate architecture of the canopy [4], and the adaptability of the rootstock [5]. Citrus rootstocks have different responses to water

stress, using different physiological strategies such as hydraulic redistribution, osmotic adjustment, and adjustment in stomatal opening [5].

Additionally, several authors have observed that under water-stress conditions, osmotic adjustment is a habitual physiological response in different plant species, whose mechanism consists in synthesizing osmoprotective substances such as proline [2]. In citrus, the osmotic adjustment occurs mainly in the root, finding accumulated inorganic solutes, such as soluble sugars, and organic solutes such as proline and glycine betaine [5].

Among the techniques and/or technologies used for increasing water productivity in citrus we can point out the use of biostimulants, irrigation strategies for reducing the applied water, and physical barriers for reducing crop evapotranspiration. In the case of biostimulant application, products are mainly foliar-applied and have a diverse origin, usually coming from different organic sources, such as seaweed extract [6,7]. These applications can increase the vigor of plants and increase the capacity to tolerate stresses with better use of water resources [6].

Irrigation strategies such as controlled deficit irrigation (CDI) and partial rootzone drying (PRD) have been evaluated in citrus, and in both cases, the results showed water saving but a significant decrease in yield and fruit quality [8]. In terms of the use of physical barriers to prevent water evapotranspiration, an example is the use of organic or plastic mulch over the soil to reduce water evaporation. In Eureka Lemon, the use of black polyethylene mulch showed a significant increase in soil moisture, improving plant growth and yield [9].

Some of the above-mentioned technologies and/or techniques have been demonstrated to significantly reduce crop water consumption. However, they involve high investments, and in some cases, negative effects on the production and/or fruit quality. Considering that in the fruit industry water saving without causing water stress in the crop is a big concern, other tools are continuously being sought. A tool of growing interest in agriculture to be used as a water-saving tool has been the use of water-retaining polymers called hydrogels, which have been reported by some authors as a clean and efficient alternative for retaining water in soil or substrate [10,11].

Hydrogels, hydro-retainers, or super-absorbent gels are hydrophilic acrylamide-based polymers with a three-dimensional structure, generally made up of long-chain, high-molecular-weight organic molecules, linked by cross-links between the chains [12]. Poly-acrylamides have the property of being highly absorbent, with a storage capacity ranging from 400 to 1500 g of water per gram of product, which improves the absorption and retention capacity of water in the soil without affecting its availability to plants [13,14]. The authors of [10] observed that the highest percentage of water absorption by potassium polyacrylamide was in soils with a sandy texture, while the authors of [12], evaluated the effectiveness of sodium polyacrylate in soil water retention, concluding that it improved the water-retention capacity of different soil textures, promoting greater efficiency in the use of both irrigation and rainfall water by reducing percolation losses. On the other hand, acrylamide is considered a toxic element, and therefore it could contaminate soil, water, and food [15]. Another long-term negative effect of these gels is by altering the physiological activities of the plants through both the toxic effect of acrylamide and a physical negative effect in soil, clogging of pores due to their high viscosity and molecular weight [15].

Although the use of polyacrylamide-based water-retaining gels has been incorporated into crop management for several species, there is little information regarding their effect on water retention by soils or substrates and the effect on the physiology of agricultural species such as citrus. Considering that hydrogels could be an alternative tool for water saving in citrus orchards, in this study, the use of polyacrylamide gels in mandarin W. Murcott under pot conditions was evaluated, to determine their effect on plant water status, biomass, and nutritional effects when applied in water-restriction conditions.

## 2. Materials and Methods

### 2.1. Experimental Setup

The experiment was conducted under climate-controlled greenhouse conditions (the daily temperature fluctuated between 15 and 32 °C with a relative humidity between 65 and 75%), at the Facultad de Agronomía e Ingeniería Forestal, Central Zone of Chile (34°07′55″ S and 70°43′15″ W).

The plant material was a 2-year-old Tangor (*Citrus sinensis* × *Citrus reticulata*), cv. W. Murcott, grafted on clonal rootstock C35 Citrange, which was established individually in plastic containers of 20 L, containing compost as substrate. The substrate chemical characteristics were as follows: 20% organic matter content (Walkley–Black method); pH 7.2 (soil: water, 1:2.5); 45 mg $kg^1$ P (Olsen method); 258 mg $kg^{-1}$ exchangeable K (ammonium acetate method); and adequate secondaries macronutrient and micronutrients. To keep similar evapotranspiration between treatments, a pruning was performed to homogenize the leaf area in a range of 1291 and 1306 $cm^2$.

The polymer used for this experiment was a sodium anionic polyacrylamide, with a pH of 8.2, and a sodium and $HCO_3$ content that corresponds to the copolymerization of acrylamide with sodium acrylate.

### 2.2. Experimental Design

The trial was conducted with a completely randomized design, with three treatments and four replications. The experimental unit was a potted plant. Treatments were applied as follows: Control (T0): irrigation with 100% of water replenishment according to crop evapotranspiration (ETc). ETc was calculated from the water-balance equation: I + PP = ETc + Pc + $\Delta\theta v$, in which I was irrigation (mm), PP was precipitation (mm), Pc was percolation (mm), and $\Delta\theta v$ corresponds to the difference of the volumetric soil moisture estimated through the daily weights of the pots. Treatment 1 (T1): 50% of water replenishment concerning Control. Treatment 2 (T2): 50% of water replenishment concerning Control plus the application of polyacrylamide polymers mixed in the substrate. To mix the polymer with the substrate, 40 g of polymer was hydrated in four liters of distilled water for one hour. This was then mixed with the substrate according to the protocol recommended by the manufacturer.

At the beginning of the experiment, all the plants were irrigated with the same amount of water until pot capacity was reached (−1 KPa of soil water tension). The frequency of irrigation varied according to the daily evapotranspiration of the control treatment, maintaining the water content of the T0 substrate near the pot capacity. To keep T0 at pot capacity, the irrigation moment was determined according to the matric potential measured with a tensiometer; when soil matric potential in T1 showed less than −1 KPa, irrigation was performed by replenishing the water volume according to the treatment. Water replenishment in the treatments was applied by keeping the same time and frequency of irrigation but using different flow drippers: 4 L/h for T0 and 2 L/h for T1 and T2. The total water volume applied for each plant during the study period was 24,6 L for T0, whereas, for T1 and T2, 12,3 L was applied.

### 2.3. Measurements

To know the atmospheric water demand, the vapor pressure deficit (VPD) was registered. VPD was determined by the equation: $VPD = 0.61078exp\,[(17.269 \times T)/(T + 237.3) \times (1 - HR/100)]$. Temperature (°C) and relative humidity HR (%) were recorded every 15 min with a Hobo Pro V2 Logger sensor.

#### 2.3.1. Substrate Temperature

The substrate temperature was evaluated every 15 days. This measurement was made at a depth of 15 cm, with a thermometer RTD Thermometer, model 505 (CHY, Taiwan).

### 2.3.2. Soil Water Content

The soil volumetric water content (θ) (substrate in this case) was measured in all the potted plants at a depth of 20 cm every 7 days using a portable Frequency Domain Reflectometry (FDR) sensor (GS-1, Decagon Devices Inc., Pullman WA 00163, USA), whose equation to estimate θ in substrate is: $θ = 4.33 \times 10^{-4} \times RAW − 0.611$, where RAW is the readily available water calculated from raw dielectric permittivity values that the device measures, with the Topp equation [16]. Prior to the measurement, the real θ of the substrate was obtained from the gravimetric method [17] and substrate bulk density [18]. With real θ values and estimated θ obtained from eight in situ measurements, a calibration curve was performed.

### 2.3.3. Plant Water Status Measurements

Midday stem water potential (SWP) was measured using the method described by the authors of [19]. Midday SWP was measured monthly in one shoot per plant between 11:00 and 15:00 h. Before each SWP measurement, the shoot was enclosed in a plastic bag covered with aluminum foil for 30 min. The shoot was then excised and SWP was measured with a Scholander pressure chamber.

Leaf relative water content (RWC) was measured between 11:00 and 15:00 h every 30 days according to [20].

### 2.3.4. Physiological Parameters

Chlorophyll fluorescence (Fv/Fm) was measured as described by the authors of [21] using a chlorophyll fluorometer (Pocket PEA, Hansatech, Norfolk, UK) and stomatal conductance (gs) was evaluated with a Leaf porometer SC-1 (Decagon Devices Inc, Pullman, WA 00163, USA). Measurements were made every 30 days between 11:00 and 15:00 h.

### 2.3.5. Growth and Biomass of Plants

At the end of the experiment, the plants were harvested and separated into leaves, shoots, and roots. Fresh weight (FW) was recorded for each plant tissue. Vegetal samples from leaves, shoots, and roots were taken and oven-dried at 65 °C to obtain a constant weight, obtaining dry matter content.

### 2.3.6. Nutrition and Metabolites Analysis

Since drought reduces the absorption and transport of nutrients from the roots to shoots, each plant tissue was nutritionally analyzed for nitrogen (N), phosphorous (P), potassium (K), calcium (Ca), magnesium (Mg), and sodium (Na).

Dry samples were subsequently ground and analyzed to determine the total N concentration using a LECO CNS-2000 Macro Elemental Analyzer (Leco, MI, USA) in the Analytical Laboratory of the Pontificia Universidad Católica de Chile.

For Ca, K, Mg, P, and Na, ashed tissue samples were then dissolved in HCl (2 M), and concentrations were determined by inductively coupled plasma–optical emission spectroscopy (ICP–OES) (Agilent 720 ES axial—Varian, Victoria, Australia).

Additionally, the root proline concentration was analyzed as an indicator of water stress, through the protocol described by the authors of [22].

### 2.3.7. Statistical Analysis

Data analysis was undertaken by one-way analysis of variance (ANOVA) and Tukey's studentized range test at $p \leq 0.05$, by using SAS statistical software (SAS Institute, Cary, NC, USA).

## 3. Results

### 3.1. Substrate and Plant Water Status

The substrate temperature ranged from 15.2 to 20.3 °C during the study period, without differences between treatments (data not shown). On the other hand, the moisture

content θ (m³ m⁻³) in the substrate showed significant differences between treatments. From May to September, the θ (m³ m⁻³) of T2 was significantly higher than T0 and T1, despite receiving the same amount of water as T1 and 50% less water than T0 (Figure 1). During October and November, θ values for T2 were similar to T0 and significantly higher than T1. It should be noted that the calibration curve between the real and estimated moisture values gave a linear direct relationship, with an R² of 0.93 with an equation that expresses the following: FDR θ = 2.4516 × Real θ − 1.64 (data not shown).

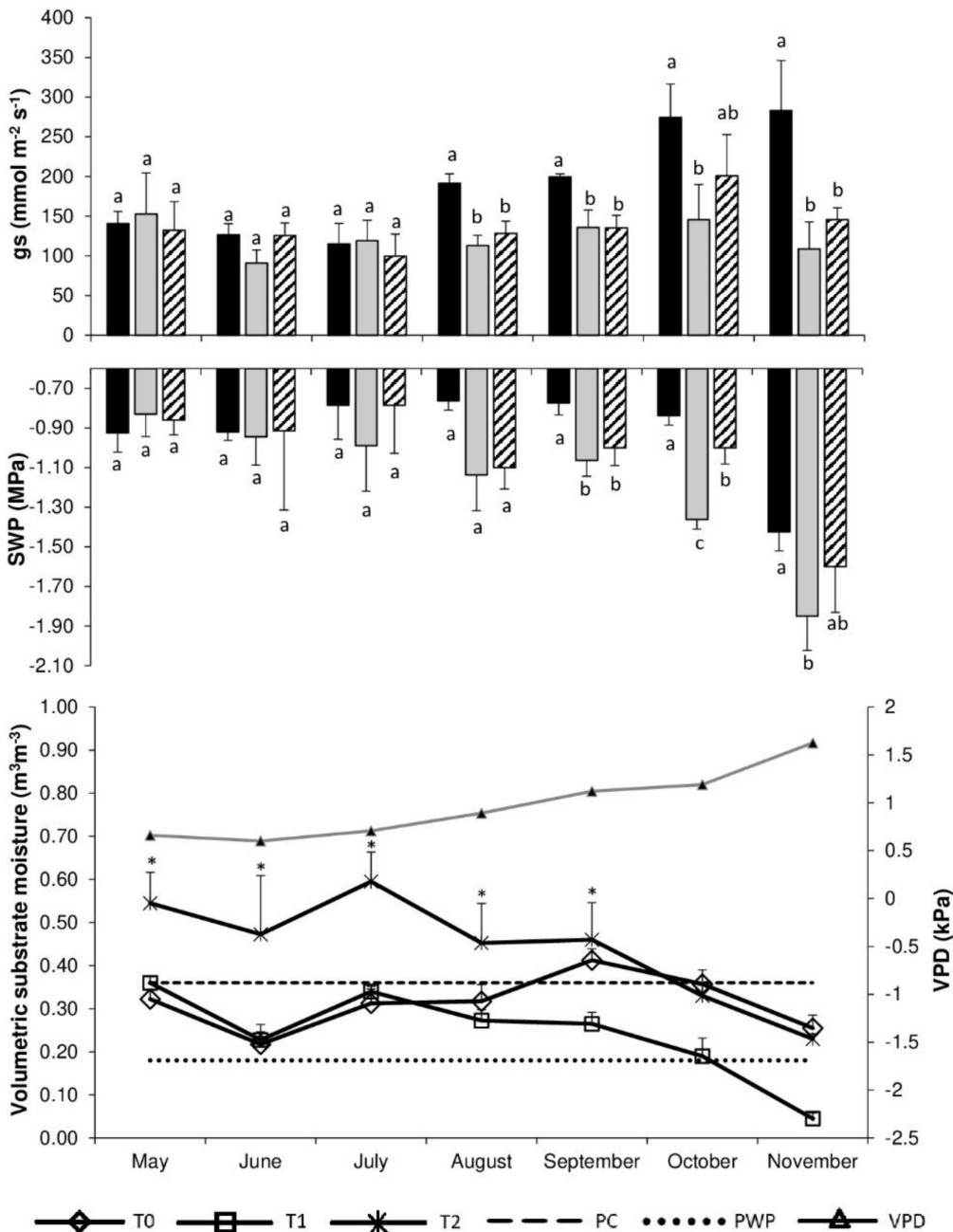

**Figure 1.** Stem-water potential (SWP), stomatal conductance (gs), and substrate moisture response to treatments: T0 (Control, 100% ETc water replenishment), T1 (50% of ETc water replenishment), and T2 (50% of ETc water replenishment + hydrogel). PC (pot capacity), PWP (permanent wilting point), and VPD (vapor pressure deficit). Different letters or asterisks indicate statistical differences (*p* ≤ 0.05, Tukey test).

The difference in moisture content θ ($m^3\ m^{-3}$) in the substrate between treatments was not reflected in physiological parameters during the first 3 months. No significant differences in gs and SWP were observed between the treatments during May, June, and July (autumn and winter months). On the contrary, from August (the end of the winter season in SH), it was possible to observe a negative effect of treatments T1 and T2 (50% water restriction) on the physiological response of the citrus plants, both in gs and SWP. In the case of fluorescence (Fv/Fm) and leaf RWC, plants did not show significant differences between the treatments (data not shown).

### 3.2. Growth and Biomass of Plants

The total biomass and shoot biomass of T0 plants were higher than the biomass of the T1 and T2 treatments. For roots and leaf biomass, no statistical differences were observed (Figure 2).

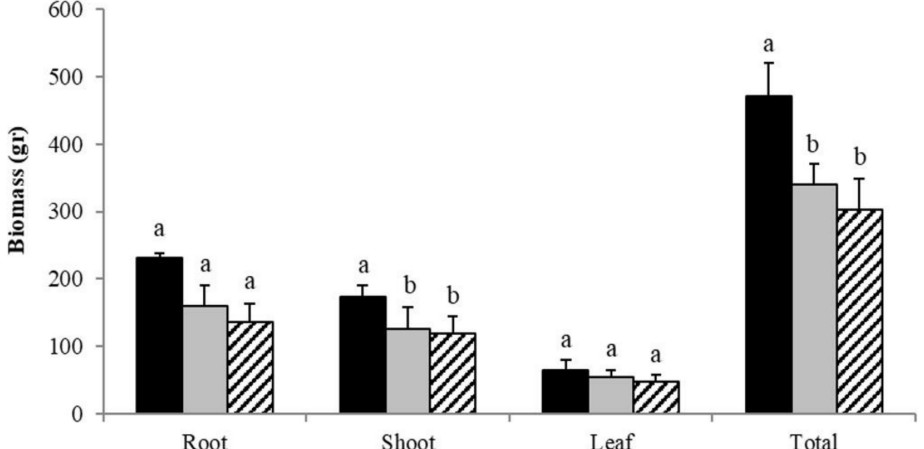

**Figure 2.** Total root, shoot, and leaf final biomass in W. Murcott trees subjected to T0 (black color) (Control, 100% of ETc water replenishment), T1 (grey color) (50% of ETc water replenishment) and T2 (hatched) (50% of ETc water replenishment + hydrogel). Different letters indicate statistical differences ($p \leq 0.05$, Tukey test).

### 3.3. Nutrition and Metabolites Analysis

Regarding the nutrient analyses, no significant differences in N, P, Ca, or Mg were observed. However, sodium content was significantly higher for T2 both in root and leaf, while potassium content was significantly higher only in leaves of T2 (Table 1).

Additionally, the proline concentration was significantly higher in the root of T2 plants compared to T1 and T0 (Table 2).

**Table 1.** Nutrient concentration in plant tissues. Total root, shoot, and leaf nutrient content in W. Murcott trees subjected to T0, T1, and T2. Different letters within the column indicate statistical differences ($p \leq 0.05$, Tukey test).

| Organ Tmt | | % N | | | % P | | | % K | | | % Ca | | | % Mg | | | % Na | | | % Si | | |
|---|---|---|---|---|---|---|---|---|---|---|---|---|---|---|---|---|---|---|---|---|---|---|
| **Root** | T0 | 1.08 | ±0.10 | a | 0.09 | ±0.01 | a | 0.69 | ±0.07 | a | 0.54 | ±0.03 | a | 0.07 | ±0.01 | a | 0.03 | ±0.013 | b | 0.10 | ±0.055 | a |
| | T1 | 0.94 | ±0.21 | a | 0.10 | ±0.02 | a | 0.68 | ±0.07 | a | 0.56 | ±0.07 | a | 0.06 | ±0.01 | a | 0.03 | ±0.004 | b | 0.10 | ±0.015 | a |
| | T2 | 1.28 | ±0.20 | a | 0.10 | ±0.01 | a | 0.68 | ±0.11 | a | 0.62 | ±0.08 | a | 0.08 | ±0.02 | a | 0.05 | ±0.006 | a | 0.12 | ±0.033 | a |
| **Shoot** | T0 | 0.93 | ±0.17 | a | 0.13 | ±0.02 | a | 0.90 | ±0.07 | a | 1.37 | ±0.19 | a | 0.07 | ±0.01 | a | 0.02 | ±0.004 | a | 0.06 | ±0.048 | a |
| | T1 | 0.81 | ±0.13 | a | 0.12 | ±0.02 | a | 0.89 | ±0.08 | a | 1.13 | ±0.17 | a | 0.06 | ±0.01 | a | 0.02 | ±0.004 | a | 0.05 | ±0.057 | a |
| | T2 | 1.02 | ±0.15 | a | 0.14 | ±0.04 | a | 0.96 | ±0.19 | a | 1.19 | ±0.18 | a | 0.07 | ±0.02 | a | 0.03 | ±0.008 | a | 0.05 | ±0.050 | a |
| **Leaf** | T0 | 2.74 | ±0.58 | a | 0.18 | ±0.02 | a | 2.46 | ±0.25 | b | 2.04 | ±0.14 | a | 0.28 | ±0.03 | a | 0.03 | ±0.003 | b | 0.012 | ±0.006 | a |
| | T1 | 2.55 | ±0.42 | a | 0.21 | ±0.03 | a | 2.62 | ±0.34 | ab | 2.10 | ±0.20 | a | 0.29 | ±0.03 | a | 0.04 | ±0.004 | b | 0.004 | ±0.002 | b |
| | T2 | 3.29 | ±0.19 | a | 0.20 | ±0.02 | a | 3.08 | ±0.27 | a | 1.94 | ±0.36 | a | 0.28 | ±0.02 | a | 0.12 | ±0.068 | a | 0.003 | ±0.001 | b |

**Table 2.** Proline concentration in fine (absorbent) roots in W. Murcott trees subjected to T0, T1, and T2. Different letters within the column indicate statistical differences ($p \leq 0.05$, Tukey test).

| Treatments | Proline Fine Root (mg g$^{-1}$) |
|:---:|:---:|
| T0 | 6.54 c |
| T1 | 9.42 b |
| T2 | 13.76 a |

## 4. Discussion

The results obtained in this study indicated that the application of these water-retaining polymers effectively improved the water content in the substrate. The results showed that the substrate moisture for T2 was kept significantly higher than T0, and T1, despite receiving the same irrigation rate as T1 and a half of T0; however, this higher moisture availability in the substrate of T2 was not reflected in the plant's water status or growth. The higher water retention observed in the substrate with polyacrylamide gels was consistent with the information reported by the authors of [13], who pointed out that polyacrylamide-based polymers are highly absorbent and insoluble in water, a characteristic that allows one to increase the water-retention capacity in the soil and/or substrate. However, the difference observed in the water retention was not consistently reflected in the physiological parameters and biomass evaluated in the plants. It was only possible to observe the effect of 50% water restriction (T1 and T2) on the physiological response of the plants, expressing themselves through mechanisms such as gs and SWP reduction [2], from August on. Other physiological responses such as chlorophyll fluorescence (Fv/Fm), indicated that the plants did not present severe stress limiting photosynthetic function (no significant differences between the treatments were observed) [23]; however, evident effects on growth and nutritional content were observed.

All these results do not coincide with what was observed by the authors of [10,12,13], who indicated that polymers favor the development of the crop. In our study, the hydrogel used demonstrated effective water retention, which is evidenced in the higher water content of the substrate compared to T1 and even T0 in some cases. If this retention capacity finally allows increases in the water available for the plants, T2 should have shown at least a greater growth than T1. However, it has no difference in biomass and has a smaller root compared to T1 and T0. The above-mentioned could indicate that the polymer has a high retention capacity but does not have a high capacity to deliver water to the plants as is required. It is probable that the observed behavior of this specific polyacrylamide gel may be given by its formulation.

A lower total biomass and root biomass for water restriction treatments, T1 and T2, was observed (Figure 2). An excess of moisture in the substrate could have reduced oxygenation, affecting the development of the roots in W. Murcott plants. This response was observed in the orange cv. Valencia [24]. On the other hand, the lower root development in T2 plants could be attributable to a lower root growth expression to seek water. Roots under little water availability respond through hydrotropism, a mechanism that modifies its growth, responding to a potential water gradient in the soil by growing towards areas with higher moisture content [25]. In this way, the lower growth of W. Murcott in the substrate with polyacrylamide hydrogel could be given by the high moisture presented in the substrate, making it unnecessary for the roots to search for areas with higher moisture. Additionally, citrus plants could have diverted photosynthates for the production of protective metabolites against saline or water stress (proline for example) instead of producing root biomass.

Nutritional analysis of the roots and leaves showed significantly higher Na content for T2. Authors such [12] observed similar results when using polyacrylamide polymers, obtaining a significant increase in sodium levels in crops such as wheat. From our results, it could be concluded that polyacrylamide hydrogels with high content of sodium could generate an accumulation of sodium in the rhizosphere, causing sodium absorption by

the roots and osmotic stress to the plants, which can cause a reduction in growth and/or osmotic adjustment if the species have the gene pool for expressing that strategy.

Proline is considered an osmotic stress indicator (water or salinity stress), as this osmolyte is commonly accumulated under abiotic stress conditions [2]. The authors of [26] indicate that citrus species under these stress conditions can respond through an osmotic adjustment, which means that cells respond by promoting the accumulation of compatible solutes (such as proline and others) to keep cellular functioning. However, this survival strategy has a negative effect on the growth of plants, since most of the energy is channeled towards the synthesis of these compatible solutes [27]. According to our results, proline concentration (mg g$^{-1}$) was significantly higher in T2 plants compared to those in T0 and T1, which may reflect an osmotic adjustment response of T2 plants, probably in response to the sodium content of the applied polyacrylamide, which probably generated saline stress. This possible adjustment response could also explain the lower growth of the shoots of the T2.

Although the addition of polyacrylamide polymers in this experiment significantly improved the retention of water in the substrate, it is questionable how much of this water was available to the W. Murcott plants. Our results showed that the plants expressed some signs of water and/or salt stress, which can be associated with a lack of water combined with a possible osmotic adjustment given by a high concentration of sodium, which leads to an increase in the concentration of osmolytes such as proline.

## 5. Conclusions

In conclusion, polymers such as sodium polyacrylamide allow water retention in the soil or substrate, but this is not necessarily released to plants. This can be explained insofar as that moisture is probably kept in the hydrogel and not released to the substrate media or the roots, or if released, in this case, this occurs with an increase in the concentration of sodium available to the plants, which could lead to the crop to a high-stress situation when the plant is subjected to water scarcity and increased growth and productivity problems. Our results also suggest that it is necessary to know the hydrogel-type product well and test it before applying it to commercial plantations.

**Author Contributions:** Conceptualization, P.M.G., C.B. and J.M.; methodology, D.C., C.B. and P.M.G.; software, D.C.; validation, D.C., C.B., J.M. and P.M.G.; formal analysis, D.C.; investigation, D.C.; resources, P.M.G. and C.B.; data curation, D.C. and P.M.G.; writing—original draft preparation, D.C.; writing—review and editing, P.M.G., C.B. and J.M.; visualization, D.C.; supervision, P.M.G.; project administration, P.M.G.; funding acquisition, D.C. and P.M.G. All authors have read and agreed to the published version of the manuscript.

**Funding:** This research received no external funding.

**Data Availability Statement:** Not applicable.

**Acknowledgments:** This work was supported by resources from the Laboratory of Water and Irrigation, of the Department of Fruit production and Oenology, Facultad de Agronomía e Ingeniería Forestal, Pontificia Universidad Católica de Chile. We thank Alejandro Campero, for contributing information and materials to this research.

**Conflicts of Interest:** The authors declare no conflict of interest.

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
