# Peer review of "Response of Potted Citrus Trees Subjected to Water Deficit Irrigation with the Application of Superabsorbent Polyacrylamide Polymers"

_agronomy, doi:10.3390/agronomy12071546_

Round 1

Reviewer 1 Report

1. Abstract

The Abstract determines the manuscript’s content and objectives in a very manifest and complete fashion.

2. Introduction

Well written.

3. Materials and Methods

The materials and methods are good enough

4. Results

Analyzed quite exhaustively but there are some comments as follow:

>> Please check and revise the Y-axis of Figure 1. "Volumetric Sustrate Moinsture"

>> The Authors mention that “the substrate moisture availability was not reflected in the plants (P:1; L: 17-18). Based on that, may we know the plant growth (height) refers to the volumetric substrate moisture as showed in the Figure 1?

5. Discussion

Quite complete.

6. Conclusions

Supported by the previous sections.

Author Response

1) Reviewer observation: Please check and revise the Y-axis of Figure 1. "Volumetric Sustrate Moinsture"

Response: The figure was revised, and the Y-axis title was corrected to “Volumetric Substrate Moisture”. Please see Figure 1 in the revised version of the Manuscript.

2) Reviewer observation: The Authors mention that “the substrate moisture availability was not reflected in the plants (P:1; L: 17-18). Based on that, may we know the plant growth (height) refers to the volumetric substrate moisture as showed in the Figure 1?

Response: We completed the idea in the abstract and now the idea is better understood. Now the paragraph says: “ The results showed that the substrate moisture for T2 was kept significantly higher than T0, and T1, despite receiving the same irrigation rate as T1 and a half of T0; however, this higher moisture availability in the substrate of T2 was not reflected in the plant’s water status or growth; on the contrary, the T2 plants showed responses such as lower total biomass, lower vegetative development, and lower root biomass, as well as a higher concentration of proline in the root. According to these results, it is concluded that polymers such as polyacrylamide sodium allow the retention of water in the substrate, but do not necessarily release that water for plants, probably because that moisture is kept in the hydrogel and not released to the substrate media or the roots, or if released, in this case, this occurs with an increase in the concentration of sodium available to the plants, which could lead the crop to a situation of water and/or osmotic stress to the detriment of growth and productivity“. Please see lines 10-27 of the corrected document.

Reviewer 2 Report

(1) the general objective and specific objectives of this study are not clearly defined

2) the research and experiments are not properly designed, the results obtained are somewhat incidental

3) what calibration equation was used in the measuring device to measure the moisture content of the soil substrate and not of the soil ?

4) was verification of the obtained measurements of soil substrate moisture carried out, e.g. using the gravimetric method ?

5) were the measurements of particular parameters, e.g. moisture content, made very rarely, e.g. once a month?

6) changes in the content of particular elements in leaves, roots did not show too big differences in the case of particular varieties of the applied research?

7) were the plants irrigated during the whole vegetation period, what were the water doses?

8) the literature review and discussion of the results were based on a relatively small amount of cited literature

9) what is new about this type of research, is that the applied substance increases the water content in the soil substrate ?

10) the research and its presentation have the character of a technical report, not a scientific article

I do not recommend it for publication in the journal Agronomy, maybe after numerous corrections and additions to a journal of local scale

Author Response

Responses to Reviewer 2

We greatly appreciate Reviewer´s 2 comments and corrections, there were very helpful to improve the quality of this research article.

1) Reviewer observation: The general objective and specific objectives of this study are not clearly defined.

Response: The abstract was corrected in order to make clear the general objective (please see lines 10-13). In the introduction the objective was better completed and now is well justified and defined (see lines 92- 99 in the corrected Manuscript).

2) Reviewer observation: The research and experiments are not properly designed, the results obtained are somewhat incidental

Response: The authors do not agree with this comment. The experiment was designed according to the proposed objectives and suggested hypothesis. The experimental design, experimental unit, number of replications, treatments, evaluations, and data analysis are clearly indicated in the methodology. The results respond to the research questions based on evidence.

3) Reviewer observation: What calibration equation was used in the measuring device to measure the moisture content of the soil substrate and not of the soil?

Response: Details of the calculations and calibration equations were included in order to make this point clearer. Please see details on lines 157-164 and 216-218 in the revised Manuscript.

4) Reviewer observation: Was verification of the obtained measurements of soil substrate moisture carried out, e.g. using the gravimetric method?

Response: The verification of obtained soil moisture reading by FDR probes was verified in comparison to real volumetric water content by a calibration curve. Details of that calibration and results are shown now in lines 162-164 and 216-218 of the revised Manuscript.

5) Reviewer observation: Were the measurements of particular parameters, e.g. moisture content, made very rarely, e.g. once a month?

Response: The measurement frequencies were well described in the methodology of the original article. In the case of substrate moisture content, the methodology described: “The soil volumetric water content (Ï´) (substrate in this case) was measured at a depth of 20 cm, every 7 days using a portable Frequency Domaine Reflectometry (FDR) sensor (GS-1, Decagon Devices).”. We improved the methodology description, and more information was added. Please see lines 157-164 of the revised Manuscript.

6) Reviewer observation: Changes in the content of particular elements in leaves, roots did not show too big differences in the case of particular varieties of the applied research?

Response: In the results section we indicate: “About the nutrient analyses, no significant differences in N, P, Ca, and Mg were observed. However, sodium content was significantly higher for T2 both in root and leaf, while potassium content was significantly higher only in leaves of T2 (Table 1).”  Please see lines 244-250 and Table 1 where is possible to see the levels of nutrients in all the sampled organs. For micronutrients, a little difference can be a big difference in terms of plant effect.

7) Reviewer observation: Were the plants irrigated during the whole vegetation period, what were the water doses?

Response: The irrigation management for treatments was well described in the methodology of the original article. “At the beginning of the experiment, all the plants were irrigated with the same amount of water until reaching pot capacity (-1 KPa of soil water tension). The frequency of irrigation varied according to the daily evapotranspiration of the control treatment, maintaining the water content of the T0 substrate near the pot capacity. To keep T0 at pot capacity, the irrigation moment was determined according to the matric potential measured with a tensiometer; when soil matric potential in T1 showed less than -1KPa, irrigation was done replenishing water volume according to the treatment. Water replenishment in the treatments was applied by keeping the same time and frequency of irrigation but using different flow drippers, 4 L/h for T0 and 2 L/h for T1 and T2 “.

In order to make this idea clear, we add more information:

“…The total water volume applied for each plant during the study period was 24,6 L for T0, whereas, for T1 and T2, 14,3 L per plant was applied.”. Please see lines 141-143.

8) Reviewer observation: The literature review and discussion of the results were based on a relatively small amount of cited literature

Response: We include 6 new references in the new version of the manuscript. Please see references 11, 14, 16, 17, 18, and 20.

9) Reviewer observation: What is new about this type of research, is that the applied substance increases the water content in the soil substrate?

Response: The novelty of this type of research is now well defined in the introduction. We include “…Although the use of polyacrylamide-based water-retaining gels has been incorporated into crop management for several species, there is little information regarding their effect on water retention by soils or substrates and regarding the effect on the physiology of agricultural species such as citrus”. Please see lines 92-95 of the revised Manuscript.

10) Reviewer observation: The research and its presentation have the character of a technical report, not a scientific article.

Response: the authors disagree with this comment. The scientific method used, the results obtained, and the discussion corresponds to a scientific article and not a technical one as the reviewer suggests. This is a research work that suggests a hypothesis and was directed to a very specific objective that responds well to the research methodology carried out.

Round 2

Reviewer 2 Report

I see the better level of this paper, I don't agree with the answer for question 4, the indirect measerements of moisture conten should be sometimes measured by gravimetric method in order to compare the results obtained by the first method. The Autors in many papers don't rememer about it.